# Degenerative Joint Damage Is Not a Risk Factor for Streptococcal Sepsis and Septic Arthritis in Mice

**DOI:** 10.3390/life11080794

**Published:** 2021-08-05

**Authors:** Johann Volzke, Brigitte Müller-Hilke

**Affiliations:** Core Facility for Cell Sorting and Cell Analysis, University Medical Center Rostock, 18057 Rostock, Germany; brigitte.mueller-hilke@med.uni-rostock.de

**Keywords:** osteoarthritis, septic arthritis, Group A Streptococcus, sepsis, anterior cruciate ligament transection

## Abstract

Septic arthritis (SA) is an aggressive joint disorder causing invalidity and mortality. Although epidemiological studies suggest osteoarthritis (OA) as a risk factor for SA, experimental insights into the relatedness of both diseases are lacking. We therefore sought to investigate whether pre-existing OA indeed promotes SA frequency or severity. We used STR/ort mice that spontaneously develop OA and, in addition, induced OA via anterior cruciate ligament transection (ACLT) in C57BL/6J mice. Mice were infected with Group A Streptococcus (GAS) and then were monitored for clinical signs of sepsis and SA. Sepsis was confirmed via elevated inflammatory cytokines in plasma, while bone morphology was assessed by micro-computed tomography. Cartilage integrity was evaluated histologically. Mice with spontaneous OA developed life-threatening SA, with GAS only moderately affecting the femoral bone structure. Surgically induced OA neither impacted on SA incidence nor on mortality when compared to infected mice without the preceding joint disease. Furthermore, only insignificant differences in bone morphology were detected between both groups. Our data indicate that degenerative joint damage due to ACLT, by itself, does not predispose mice to SA. Hence, we propose that other factors such as prosthetic joint replacement or high age, which frequently coincide with OA, pose a risk for SA development.

## 1. Introduction

Septic arthritis (SA) is a life-threatening disorder that is primarily triggered by Gram-positive bacteria invading the affected joint [1]. Upon infection of the articular microenvironment, pathogens rapidly propagate and trigger a local inflammatory immune response. In the event of the host failing to eradicate the pathogen, the acute inflammation causes irreversible damage of bone and cartilage within a few days of disease onset [2]. About 30% of patients surviving SA suffer from chronic damage to the locomotor system, rendering them disabled for life [3]. While mortality rates of SA remain continuously high, experimental models describing the pathogenesis of this rheumatic emergency are scarce [4].

Epidemiological data suggest that patients with pre-existing joint disorders such as rheumatoid arthritis (RA) or osteoarthritis (OA) are at a high risk of developing SA [4,5]. We recently demonstrated that collagen-induced arthritis—a mouse model for RA—promoted the onset and severity of SA, which, in turn, was associated with immune dysregulation and increased mortality [6]. Using mouse models for human OA, we aimed to investigate if degenerative joint damage, by itself, also increases the risk for SA. To that extent, we here compared primary and secondary OA after infection with Group A Streptococcus (GAS) which is an important causative agent of SA [7]. Septic arthritis was clinically evaluated, and we performed bone morphological, histological and serological analyses to investigate possible correlations between different disease entities.

## 2. Materials and Methods

### 2.1. Small Animal Models for Primary and Secondary Osteoarthritis

C57BL/6J and STR/ort mice were obtained from Charles River and Harlan Winkelmann, respectively. Mice were kept at 21 ± 2 °C ambient temperature in individually ventilated cages with water and food given ad libitum. A total of 48 animals were used for the experiments. Mice from different litters were allocated to groups in order to minimize confounders. All investigators were aware of the allocations at all points of the experiments. Male STR/ort mice are known to spontaneously develop erosions of knee and ankle joints beginning from an age of 18 weeks [8]. Hence, 20- to 22-week-old STR/ort mice served as a model for primary OA. Secondary OA was established by surgically destabilizing the right knee joint via anterior cruciate ligament transection (ACLT) in 6- to 8-week-old male C57BL/6J mice [9]. In brief, medial arthrotomy and removal of the infrapatellar fat pad exposed the anterior cruciate ligament which was transected before the joint capsule and skin were sutured (Appendix A). In order to control for adverse events during and after the surgery, sham surgery was performed where the joint capsule was sutured after removal of the infrapatellar fat pad without transecting the ACL. Metamizole was administered peri-surgically to all mice until three days after the procedure. In this model for secondary OA, radiologically detectable joint damage occurs within several weeks after surgery [9]. Therefore, GAS infection of C57BL/6J mice was performed 14 weeks after ACLT at an age of 20 to 22 weeks.

### 2.2. GAS Culture, Infection and Clinical Scoring

Culture of GAS was performed as previously reported [6]. In short, bacteria were cultured in Todd Hewitt broth (Becton Dickinson) until mid-log phase. C57BL/6J and STR/ort mice were infected with 0.5 × 10^6^ and 1.5 × 10^6^ colony-forming units, respectively, by injecting 100 µL GAS suspension into the lateral tail vein. PBS was injected in the same fashion which served as the vehicle control. Mice from all groups were administered tramadol to reduce pain immediately before infection and were monitored for a maximum of 14 days post-infection. A sepsis scoring system was used to detect humane endpoints, as previously described in an unblinded manner [6]. Activity of (septic) arthritis was scored according to a scheme that was previously described [6,10]: 1–5 points for swollen digits, 5 points for swollen metacarpus or metatarsus, 5 points for swollen ankle or wrist joints and 5 points if GAS was isolated from tibiofemoral joint swabs postmortem. Mice were sacrificed by cervical dislocation, and blood, liver and spleen were collected. Blood was used for serological and bacteriological analyses. Plasma cytokines were quantified using the LEGENDplex^TM^ Mouse Inflammation Panel (BioLegend) according to the manufacturer’s instructions. GAS-specific antibodies were detected using an in-house enzyme-linked immunosorbent assay, as previously described [6]. Liver and spleen were homogenized and plated on agar for determination of bacterial burdens. Hind paws were dissected and fixed in formaldehyde.

### 2.3. Micro-Computed Tomography and Histology

Femora were isolated and rinsed in 0.9% NaCl solution. X-ray images were acquired and evaluated using a SkyScan1076 (Bruker, Billerica, MA, USA) with the previously reported parameters [6]. The Bruker software pipeline was used for morphometry data analyses. Femora were reconstructed and spatially aligned, and reference levels were selected at the proximal and distal epiphyses (Appendix A). Built-in algorithms were utilized for the determination of morphology parameters of binarized images. Paws were decalcified, dehydrated and embedded into paraffin. Serial sections were generated, rehydrated and stained with Safranin O, Fast Green FCF and hematoxylin (Merck, Darmstadt, Germany).

### 2.4. Statistical Analysis

Data visualization and statistical analyses were performed in R (version 3.5.1). Due to the low sample sizes, data were considered to not follow a Gaussian distribution. Survival probabilities and incidences were evaluated by performing logrank tests. The Pearson product-moment correlation coefficient was used to analyze the dependency of two quantitative variables. Analyses and comparison of univariate quantitative data were performed using the Mann–Whitney *U* test. Box plots illustrate the median including the interquartile range (IQR). No inclusion or exclusion criteria were set for the respective analyses.

## 3. Results

### 3.1. GAS Induced Severe SA in Mice with Primary Osteoarthritis but Did Not Cause Bone Erosion

We sought to investigate whether GAS triggered SA in mice with established spontaneous OA. To that end, we infected STR/ort mice and monitored them for the occurrence of sepsis and paw swellings (Figure 1A). SA was confirmed by isolation of bacteria invading the tibiofemoral joints. During the observation period, we detected SA in over 65% (8/12) of infected mice (Figure 1B, left panel). Interestingly, all animals suffering from SA had to be sacrificed within one to four days after GAS infection due to severely progressing disease. Consequently, the survival rate of all GAS-infected mice was below 35% (4/12, Figure 1B, right panel). We measured GAS-specific antibodies in the plasma from mice surviving until day 14 post-infection (median absorbance A450 = 0.234 [IQR 0.154–0.328] in the GAS group, with *n* = 4, versus A450 = 0.023 [0.008–0.054] in the PBS group, with *n* = 7, *p* = 0.0061). Of note, we detected a strong correlation of plasma concentrations of interleukin (IL-)6, interferon (IFN)γ and tumor necrosis factor (TNF)α with sepsis activity, and, indeed, the highest cytokine levels were found in mice suffering from SA (Figure 1C). Micro-computed tomography (µCT) analyses of femoral epiphyses revealed only insignificant GAS-induced changes to the morphology of the cortical bone (Figure 1D). However, we detected an unexpected increase in the bone volume fraction (BV/TV), bone surface density (BS/TV) and trabecular numbers (Tb.N) for the cancellous bone of infected mice (Figure 1D, Appendix A).

In summary, our data show that GAS infection of mice with primary OA led to a high incidence of SA, which was accompanied with a low survival probability. Yet, we did not detect marked differences in bone morphology between infected and non-infected STR/ort mice.

### 3.2. Degenerative Knee Joint Damage Did Not Predispose Mice to Septic Arthritis

In order to investigate whether existing joint damage promotes SA onset and severity, we combined a mouse model of secondary OA with GAS infection (Figure 2A). To that extent, we performed an ACLT on the right knee joint of C57BL/6J mice, and in order to ensure the establishment of degenerative changes to the joints, animals were left to sit for 14 weeks. The presence of OA was confirmed postmortem by bone morphometric analyses of the distal femoral epiphyses using µCT. Figure 2B and Appendix A show representative morphology parameters demonstrating that OA impacted on the cancellous and cortical bones alike. BV/TV and Tb.N were reduced and paralleled by a decrease in cortical thickness (Ct.Th) and a slight reduction in the cortical area fraction (Ct.Ar/Tt.Ar, *p* = 0.32). Moreover, we found a decreased eccentricity (ε) of the cortical bone in mice that underwent ACLT as well as a decreased bone surface density (BS/TV), whereas most of the parameters analyzed only changed slightly (Appendix A).

SA was triggered by GAS infection of mice with (GAS + OA) and without OA (GAS). Subsequently, mice were monitored for two weeks (Figure 2A). SA occurrence and severity were again determined via confirmation of GAS in the synovial fluid and the scoring of paw swellings. Only one out of ten animals with OA developed symptoms of SA, with GAS being isolated from the osteoarthritic joint (Figure 2C, upper panel). The SA incidence in the control group of C57BL/6 mice without OA was marginally higher (36%, 3/10, *p* = 0.22), as was the overall infection severity, as shown by a slightly increased mortality (Figure 2C, lower panel, *p* = 0.13). Likewise, bacterial burdens in the blood, liver, spleen and right knee joints were only insignificantly different between the ACLT and control groups (Appendix A). Interestingly, though, there was a strong positive correlation between sepsis scores and plasma concentrations of IL-6, IFNγ and TNFα. These correlations were independent of the preceding joint condition and confirmed a cytokine storm in severe cases of GAS infection (Figure 2D). Accordingly, there were no statistically significant differences in the plasma concentrations of IL-6 (*p* = 0.08), IFNγ (*p* = 0.20) or TNFα (*p* = 0.28, Kruskal–Wallis test) between the three treatment groups. Moreover, infected mice surviving until the end of the observation period developed GAS-specific antibodies, as shown by a median absorbance (A_450_) of 0.441 ([IQR 0.407–0.844] *n* = 12) compared to A_450_ = 0.096 in non-infected mice ([IQR 0.089–0.100] *n* = 9, *p* = 6.8 × 10^−6^).

In order to assess OA- and GAS-induced cartilage erosion, we stained histological sections of tibiofemoral joints with safranin O. The cartilage of healthy joints was characterized by a smooth surface and regular staining of proteoglycans (Appendix A, PBS). In contrast, OA led to the degradation of the extracellular matrix. In an especially severe case of SA, we detected profound deterioration of cartilage and loss of proteoglycans in an infected joint (Appendix A, GAS). Similarly, in a case of an infected osteoarthritic knee joint, we also found cartilage erosion and loss of the extracellular matrix. However, due to the low incidence of SA, these two cases of severe cartilage damage were not representative for either group of infected mice. Finally, we compared µCT data of bone morphology parameters between the GAS and the OA+GAS groups and found only minimal differences (Appendix A).

In summary, our data demonstrate that degenerative joint damage due to ACLT neither impacted on the severity of acute GAS-induced disease nor promoted the onset of septic arthritis. However, we presented two severe cases of joint infection which were associated with marked cartilage erosion.

## 4. Discussion

We have demonstrated that surgically induced degenerative joint damage after ACLT did not pose a risk for SA onset and severity. In fact, morbidity and mortality were slightly lower than in mice without the preceding joint disorder. Our results from an animal model therefore contradict epidemiological data derived from human patients which suggest OA as an important risk factor for SA [4,5]. However, data on human patients are often collected retrospectively and are likely to neglect factors other than OA that may promote the development and perpetuation of invasive infections [5].

In fact, OA is not a single disease entity but rather a combination of different pathologies interfering with the equilibrium between joint degradation and tissue repair mechanisms [11]. Arguably, degenerative joint disorders are mostly a health burden for the elderly. According to the Centers for Disease Control and Prevention, around 60% of patients at the age of 70 and above suffer from primary OA [12]. Due to age-related immune senescence and erroneous immune regulation, patients of a high age have an elevated risk of suffering from life-threatening infectious diseases such as SA [13,14]. Moreover, especially severe cases of OA require joint replacement, and a past study identified articular prosthetics as an important risk factor for SA [15]. Implant materials, due to their physical properties and their interaction with biological tissues, promote the establishment of bacterial biofilms and therefore thwart the elimination of the pathogen from the joint [16].

Despite our novel insights into the interdependencies between secondary OA and SA, there are limitations to our study. In their work, Kamekura and colleagues described that ACLT induces a mild form of OA [9]. Yet, the extent of bone lesions was measured 8 weeks after surgery, whereas mice in our study were left to sit for 14 weeks. Therefore, we cannot exclude that OA in our model was either moderate or severe. There are, however, models for secondary OA that induce fulminant joint damage, e.g., when transecting the patellar ligament [9]. Hence, although ACLT is the most frequently used model for secondary OA [17], other models for surgically induced degenerative joint disease might affect SA frequency and severity differently.

Furthermore, in this study, we used STR/ort mice as a model of primary OA. We have shown that GAS infection led to a high incidence of SA and mortality in these mice. However, we did not control for the underlying degenerative joint disease as all male STR/ort mice develop OA. Therefore, we were not able to exclude primary OA as a risk factor for SA. Past studies used the parental CBA strain, from which STR/ort mice originate, as control animals without OA [18]. However, several generations of breeding may have led to genetic variations that alter the immunological layout of STR/ort mice which might thus react differently to GAS challenge than CBA mice [19]. In summary, our results show that pre-existing degenerative joint disease due to ACLT did not predispose mice to SA, and articular damage without impairment or derailment of the immune response did not facilitate bacterial colonization of the synovia. We therefore propose that an elevated incidence of SA in OA patients may result from as yet ill-defined coinciding factors.

## Figures and Tables

**Figure 1 life-11-00794-f001:**
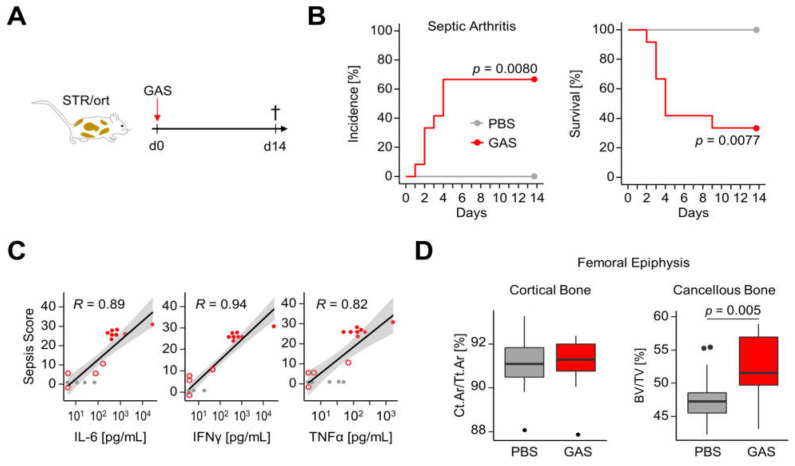
GAS induced severe disease in a mouse model of primary osteoarthritis without radiological evidence of bone erosion. (**A**) The experimental scheme is shown. STR/ort mice were infected with GAS, monitored for a maximum of two weeks and then sacrificed (^†^). (**B**) Kaplan–Meier estimator curves of septic arthritis incidence (left) and survival (right) comparing infected mice (GAS, *n* = 12 from 5 litters) to non-infected mice (PBS, *n* = 7 from 4 litters). (**C**) Dot plots including correlation analyses show the relationship of sepsis disease activity with plasma concentrations of interleukin (IL-)6, interferon (IFN)γ and tumor necrosis factor (TNF)α. Gray circles: PBS group. Open red circles: surviving mice in the GAS group without SA. Closed red circles: mice in the GAS group with SA. *R*: Pearson product-moment correlation coefficient. Gray areas show 0.95 confidence intervals of regression lines. (**D**) Representative box plots illustrate the comparisons of the femur morphometry parameters cortical area fraction (Ct.Ar/Tt.Ar) and bone volume fraction (BV/TV) of the cancellous bone between non-infected (gray boxes, *n* = 14) and infected mice (red boxes, *n* = 22). Bones from one infected animal had to be excluded from the analysis due to corruption of the raw data.

**Figure 2 life-11-00794-f002:**
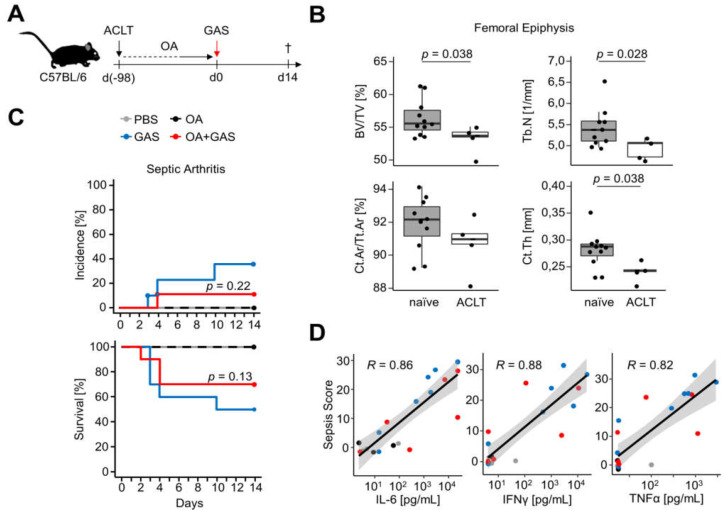
Preceding degenerative joint disease did not impact on GAS-induced sepsis and septic arthritis in a mouse model of secondary osteoarthritis. (**A**) The experimental scheme included an anterior cruciate ligament transection (ACLT) at the right knee joint of C57BL/6 mice. Subsequently, animals were left to sit for 14 weeks to ensure the development of osteoarthritis (OA). Mice were then infected with GAS and sacrificed after a maximum of 14 days after infection (†). (**B**) The establishment of OA was confirmed by comparing postmortem micro-computed tomography parameters of knee joints which underwent ACLT (*n* = 5) to joints that underwent sham surgery (naïve, *n* = 11). Bones from one treatment-naïve animal had to be excluded from the analysis due to corruption of the raw data. Representative box plots show bone volume fraction (BV/TV) and trabecular number (Tb.N) in the epiphyseal cancellous bone of the femur, as well as cortical area fraction (Ct.Ar/Tt.Ar) and cortical thickness (Ct.Th). (**C**) Kaplan–Meier estimator curves of septic arthritis incidence (top) and survival (bottom) after GAS infection. The results show a slight decrease in septic arthritis frequency and mortality when comparing infected mice with preceding OA (OA + GAS, *n* = 10 from 3 litters) to infected mice without OA (GAS, *n* = 10 from 6 litters). Non-infected animals with OA (*n* = 5 from 1 litter) and without OA (PBS, *n* = 4 from 2 litters) served as controls. (**D**) Sepsis disease activity positively correlated with the plasma concentrations of IL-6, IFNγ and TNFα. *R*: Pearson product-moment correlation coefficient. The gray areas show 0.95 confidence intervals of regression lines.

## Data Availability

The raw data supporting the conclusions of this article will be made available by the authors upon reasonable request.

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
