# Peer review of "Degenerative Joint Damage Is Not a Risk Factor for Streptococcal Sepsis and Septic Arthritis in Mice"

_life, 2021, doi:10.3390/life11080794_

Round 1
Reviewer 1 Report
Overall: The authors have used two different osteoarthritis models (primary and secondary OA) to illustrate whether GAS can lead to exacerbate significant OA disease development.
Major correction
The results have shown that mice with primary OA, but not secondary OA (i.e. ACLT-induced OA), lead to increase incidence of AS. Authors should discuss the underying mechanisms between two models being tested explaining the different outcome following GAS infection.
Additionally, the statement in line 221 should also be amended to indicate that only secondary OA did not increase SA incidence, given the fact that primary OA in mice in fact lead to a higher incidence of SA.
Minor correction
- Please edit the format in line 68 from "0.5 x 106 and 1.5x 106" to "0.5 x 106 and 1.5 x 106"
- line 152; i believe that "mice with (GAS) and wihtout OA (OA+GAS)" should read "mice with (OA+GAS) and without OA (GAS).
Author Response
Dear Reviewer,
Your comments and suggestions are very well appreciated. All the changes are implemented into the new version of the manuscript which will be uploaded shortly after this message is submitted. Please note that the manuscript contains changes in response to the other reviewer’s comments and that line numbers may have been altered. You can find my response to your suggestions below. Please also see the attachment (Supplementary Material).
Major Correction
In fact, we were not able to show that primary OA had an effect on SA incidence as there are no appropriate control mice without the pre-existing joint disease. As stated in the lines 216-220, STR/ort mice develop OA in general. However, you made a good point and the paragraph in the discussion was changed accordingly. In summary, we are now stating that – in the past – the parental CBA strain, from which STR/ort were originally obtained, was often used as a control for OA. CBA mice do not develop the joint disease. However, due to several generations of in- and cross-breeding STR/ort mice may exhibit an altered immunological layout and therefore react differently to a bacterial challenge than CBA mice.
Minor Correction
- Line 68: The numbers are now correctly formatted.
- Line 152: Nice catch! The sentence was corrected.
Kind regards,
Johann Volzke

Reviewer 2 Report
This study investigated whether degenerative joint damage due to ACL transection is a risk factor for streptococcal sepsis and septic arthritis in mice. This study compared STR/ort mice that spontaneously develop OA to C57BL/6J mice that underwent ACL transection. The authors show that STR/ort mice developed life threatening septic arthritis shortly after Group A Streptococcus (GAS) infection, while C57BL/6J mice that underwent ACL transection did not. The manuscript is well written, the methods are clear and the figures and the tables supplemented well in the text. However, a few points need to be addressed.
Comments:
- For their surgically-induced OA model, the authors refer to Ref (9) and how this particular model "represents a mild form of OA". However, the study of ref (9) only assessed up to 8 weeks post-injury. It is unclear if the "mild OA" is still present at 14-15 weeks after injury. Proof of this is necessary, i.e,. in the supplemental data Table S2 (arthritis score) and Table S3 (Bone morphometry), the authors should show the data (and p values) from the PBS and OA groups and comment on this in the Results and Discussion.
- Abstract: Line 20, should state somewhere "due to ACL transection" since different types or more severe types of joint damage could potentially predispose to SA development.
- The authors state that the fat pad was removed during the ACL transection procedure. Some studies suggest that the fat pad contributes to inflammation. Could this have an effect on the results found in the C57BL/6J mice that underwent ACL transection?
- Which arthritic scoring system was used for Table S2? Add this to methods with the appropriate citation. Is there date for the STR/ort mice on this?
- For the cytokine concentrations, were there any significant differences in concentrations of IL-6, IFNg, and TNFa (ANOVA between the 3 treatment groups)? And for the septic scores?
- Line 197 Discussion, insert ACL tear in sentence: "We have demonstrated that surgically induced degenerative joint damage did not pose a risk for SA onset and severity" since other types of surgically-induced models could pose a risk for SA onset and severity....which should be stated somewhere in Discussion.
- Discussion, line 221 (1st sentence) insert "due to ACL tear"
Author Response
Dear Reviewer,
Your comments and suggestions are very well appreciated. All the changes are implemented into the new version of the manuscript which will be uploaded shortly after this message is submitted. Please note that the manuscript contains changes in response to the other reviewer’s comment and that line numbers may have been altered. You can find my response to your suggestions below. Please also see the attachment (Supplementary Material).
Comments:
- Good catch! Thank you for pointing that out. Although Figure 2B exemplifies how ACLT impacted on bone morphology 14 weeks after surgery, we did not score or categorize OA in our model into mild or severe. Additional bone morphometry parameters were added as a table in the supplementary material (Table S2, former Table S2 is now table S3). Accordingly, the results section 3.2 has been updated and there is a paragraph about limitations within the discussion section. Finally, the phrase “mild form of OA” was removed from the methods section 2.1.
- Thanks for the suggestion. The abstract was improved upon your advice. I apologize for not being exact.
- Unfortunately the methods section 2.1 (lines 57-59) was incomplete. In fact, we performed sham surgery for mice in the PBS and GAS group, respectively. The fat pad was removed from the knee of these mice without damaging the ACL or the joint in any other way. The methods section was updated accordingly.
- You can find the scoring system for (septic) arthritis under the methods section 2.2. I apologize if the phrasing was vague. The section was improved accordingly and a citation was added from a former study that incorporated a similar scoring method for arthritis.
- In fact, there were no statistically significant differences for IL-6, IFNg or TNFa between the three treatment groups. The results section 3.2 was therefore expanded. You can find the results for sepsis scores in the table S2 (now table S3) which compares GAS with OA+GAS mice. PBS or OA(+PBS) mice did not show any signs of burden during the observation period after application of the vehicle solution.
- “ACLT” was added to the sentence in Line 197. Thank you for pointing out the limitation of our study. The possibility that other models for secondary OA may predispose to SA is now incorporated in the discussion.
- The word group was added.
Kind regards,
Johann Volzke

Round 2
Reviewer 1 Report
Many thanks to the Author to consider my suggestions.
Reviewer 2 Report
All comments have been appropriately addressed.